**Data Availability Statement:** All underlying data is included in the paper.

**Funding:** This work was supported by the National Institutes of Health under National Institute of

# The impact of vaccination to control COVID-19 burden in the United States: A simulation modeling approach

Oguzhan Alagoz[1,2]*, Ajay K. Sethi[2], Brian W. Patterson[3], Matthew Churpek[4], Ghalib Alhanaee[1], Elizabeth Scaria[1], Nasia Safdar[5,6]

**1** Department of Industrial and Systems Engineering, University of Wisconsin-Madison, Madison, Wisconsin, United States of America, **2** Department of Population Health Sciences, University of Wisconsin-Madison, Madison, Wisconsin, United States of America, **3** Berbee Walsh Department of Emergency Medicine, University of Wisconsin-Madison School of Medicine and Public Health, Madison, Wisconsin, United States of America, **4** Pulmonary and Critical Care Division in the Department of Medicine, University of Wisconsin-Madison, School of Medicine and Public Health, Madison, Wisconsin, United States of America, **5** Infectious Diseases Division in the Department of Medicine, University of Wisconsin-Madison School of Medicine and Public Health, Madison, Wisconsin, United States of America, **6** William S Middleton Memorial Veterans Hospital, Madison, Wisconsin, United States of America

* alagoz@engr.wisc.edu

## Abstract

### Introduction

Vaccination programs aim to control the COVID-19 pandemic. However, the relative impacts of vaccine coverage, effectiveness, and capacity in the context of nonpharmaceutical interventions such as mask use and physical distancing on the spread of SARS-CoV-2 are unclear. Our objective was to examine the impact of vaccination on the control of SARS-CoV-2 using our previously developed agent-based simulation model.

### Methods

We applied our agent-based model to replicate COVID-19-related events in 1) Dane County, Wisconsin; 2) Milwaukee metropolitan area, Wisconsin; 3) New York City (NYC). We evaluated the impact of vaccination considering the proportion of the population vaccinated, probability that a vaccinated individual gains immunity, vaccination capacity, and adherence to nonpharmaceutical interventions. We estimated the timing of pandemic control, defined as the date after which only a small number of new cases occur.

### Results

The timing of pandemic control depends highly on vaccination coverage, effectiveness, and adherence to nonpharmaceutical interventions. In Dane County and Milwaukee, if 50% of the population is vaccinated with a daily vaccination capacity of 0.25% of the population, vaccine effectiveness of 90%, and the adherence to nonpharmaceutical interventions is 60%, controlled spread could be achieved by June 2021 versus October 2021 in Dane County and November 2021 in Milwaukee without vaccine.

Allergy and Infectious Diseases (NIAID) Grant
1DP2AI144244-01. The funders had no role in
study design, data collection and analysis, decision
to publish, or preparation of the manuscript.

**Competing interests:** Oguzhan Alagoz is a paid
consultant for Bristol Myers Squibb and Biovector
Inc, outside of the submitted work. Other authors
have no conflicts of interest to report. This does
not alter our adherence to PLOS ONE policies on
sharing data and materials.

## Discussion

In controlling the spread of SARS-CoV-2, the impact of vaccination varies widely depending not only on effectiveness and coverage, but also concurrent adherence to nonpharmaceutical interventions.

## Introduction

With over 32 million cases in the US alone, poorly controlled transmission of SARS-CoV-2 has challenged the capacity of health systems and has resulted in over 580,000 deaths [1]. Until recently, the only effective measures for controlling the spread of SARS-CoV-2 have been non-pharmaceutical interventions (NPIs), such as physical distancing and masking. Sustained adherence to these NPIs is variable and difficult to achieve [2].

The US Food and Drug Administration authorized the use of two, two-dose vaccines and one, single-dose vaccine against SARS-CoV-2 of similarly high efficacy to prevent symptoms of novel coronavirus disease (COVID-19) and hospitalization [3, 4]. As of May 12, 2021, 36% of the US population has been fully vaccinated [1]. However, if and when widespread use of these vaccines will result in sustained control of the pandemic remains unclear given the speed of vaccine roll out and societal factors such vaccine hesitancy and suboptimal adherence to NPIs. Moreover, there is a need to examine these factors at a regional level given the rolling nature of the pandemic and COVID-19 hot spots around the country.

The objective of this study was to use our previously developed COVID-19 agent-based simulation model [2], to examine the impact of vaccine coverage and effectiveness, vaccination capacity, and adherence to NPIs on SARS-CoV-2 burden, as well as to predict how these factors influence control of virus spread in urban communities in the US.

## Methods

This study used only publicly available de-identified data. We previously developed the COVID-19 Agent-based simulation Model (COVAM) [2] to represent the interactions among people that may lead to transmission of SARS-CoV-2 in three urban regions in the US: Dane County in Wisconsin, the Milwaukee metro area in Wisconsin, and New York City (NYC). Briefly, COVAM works as follows: individuals who belong to one of eight possible states representing an individual's COVID-19-related status (**S1 Fig in S1 Data**) interact with each other through which SARS-CoV-2 is transmitted. The number of such close interactions in a given day is estimated separately for different regions considering population densities using data from social network literature and calibration [5, 6]. For each of these daily interactions, there is a possibility that a contagious individual exposes another individual to SARS-CoV-2. The number of daily contacts differs by age group, with older individuals less likely to have such interactions as compared to younger individuals. COVAM considers the possibility that not all individuals infected with SARS-CoV-2 will be tested positive and reported. Moreover, COVAM considers the possibility that some asymptomatic individuals transmit the disease prior to showing symptoms. COVAM also assumes that individuals who are experienced a previous COVID-19 infection gain protective immunity for future infections. The basic reproduction number (R0) corresponding to the unmitigated base-case transmission dynamics was 3.34 for Dane County and Milwaukee and 6.68 for NYC due to a large number of daily contacts.

A unique feature of COVAM is its ability to represent varying levels of adherence of the local communities to NPIs implemented in different regions. COVAM uses cell-phone

mobility data and calibration to estimate a time-dependent adherence to NPIs and implements it explicitly by adjusting the number of contacts per person each day [7–9]. For instance, a 70% adherence level to NPIs in Dane County and Milwaukee is implemented by simply reducing the number of daily contacts from 10 per day to 3 per day per person, which slows the transmission of SARS-CoV-2. COVAM uses adherence to NPIs to model several distinct behaviors to mitigate transmission, including mask wearing and reduced number of interactions where individuals do not maintain a 6-feet distance during person-to-person interactions, and estimate future burden of COVID-19. COVAM has the ability to represent age-specific adherence to NPIs. Similarly, COVAM allows individuals to show dynamic behavior and have varying levels of adherence to NPIs over time.

COVAM was calibrated using historical pandemic data from the three urban regions and validated in the short term with data that were not used in model development. COVAM is programmed using C++ programming language for flexibility and quicker computational times. All experiments in this study are conducted in personal computers. Additional details are available in Alagoz et al. [2] and in the Supplement.

## COVAM updates

We made two extensions to COVAM to answer the research questions of this study. The original version of COVAM used a fixed level of adherence to NPIs for future projections. However, cell-phone-based data and several studies have shown that the response of local communities to NPIs changed over time; therefore, the assumption of a fixed adherence level may not be realistic [7–10]. Specifically, even in the absence of seasonal changes, the observed number of cases and deaths impacted how local communities followed NPIs [10–12]. To this end, we added a *dynamic adherence* scenario in which we defined two thresholds that trigger high or low adherence to NPIs. Namely, if the number of daily new cases exceeds a high threshold, then the NPI adherence level increases significantly from observed levels. Similarly, if the number of daily new cases drops below a low threshold, then the adherence level drops significantly.

The second extension was the incorporation of vaccination into COVAM. We assumed that some individuals undergo vaccination and as a result may gain protective immunity and become non-susceptible after vaccination representing the *vaccine effectiveness* rate. We assumed that individuals who gain protective immunity after vaccination cannot get infected with COVID-19 and do not transmit the disease to others. We also considered that only a proportion of the population agrees to be vaccinated, and there is a daily capacity for vaccination as a function of the proportion of the population in the community. We also allowed vaccinated individuals (whether they are immunized or not) to have a lower adherence to NPIs. We did not explicitly model the timeline between first and second doses for the two-dose vaccines. Instead, we assumed that immunization is developed 14 days after the second dose of the vaccine is administered for the two-dose vaccines.

Finally, we calibrated COVAM to the latest data on the reported number of confirmed cases in Dane County, Milwaukee, and NYC using case counts by January 5, 2021, January 5, 2021, and January 14, 2021, respectively and compared COVAM's predictions against data until February 1, 2021.

## Vaccination scenarios

In all runs, we assumed that vaccination started on January 5, 2021, as the first vaccine was administered on December 14, 2020 in the US [13]. We used COVAM to evaluate the impact of different aspects of vaccination and NPI adherence using the model inputs as presented in

**Table 1. Description of vaccination scenarios.**

| Parameter | Description | Values |
|---|---|---|
| Vaccine effectiveness | Proportion of the individuals who gain protective immunity after vaccination | 50%, 75%, 90% |
| Vaccine coverage | Proportion of the population receiving full dose of the vaccines | 25%, 50%, 60%, 75%, 100% |
| Daily vaccination capacity | Proportion of the population that are vaccinated on a given day | 0.05%, 0.1%, 0.25%, 0.5% |
| Adherence to nonpharmaceutical interventions after February 1, 2021 under fixed adherence scenario | Proportion of the population following nonpharmaceutical interventions under the fixed adherence scenario | Dane County: 75%, 70%, 65%, 60% Milwaukee: 75%, 70%, 65%, 60% NYC: 90%, 85%, 80%, 75% |
| Adherence to nonpharmaceutical interventions in the future under fixed adherence scenario | Proportion of the population following nonpharmaceutical interventions under the dynamic adherence scenario | Dane County: If the number of new confirmed cases in a day is ≤50, adherence drops to 60%, if it is >50, then adherence increases to 75% Milwaukee: If the number of new confirmed cases in a day is ≤150, adherence drops to 60%, if it is >150, then adherence increases to 75% NYC: If the number of new confirmed cases in a day is ≤250, adherence drops to 75%, if it is >50, then adherence increases to 90% |
| Drop in adherence to nonpharmaceutical interventions among vaccinated individuals | Vaccinated individuals may be less likely to follow nonpharmaceutical interventions | 20%, 0% |

**Table 1**. We used observed adherence levels until February 1, 2021 and assumed that adherence levels after this date are maintained afterwards. The adherence scenarios in **Table 1** are based on the new cases and the adherence levels observed in these regions since the beginning of the pandemic (**S1 Table in S1 Data**).

We ran 100 replications for each experiment to obtain stable estimates and report only mean values due to very low standard errors. Unless noted otherwise, we reported the cumulative number of confirmed cases associated with each scenario on December 31, 2021.

Our base case scenario assumed that vaccination starts on January 5, 2021; daily vaccination capacity is 0.25% of the population per day (i.e., 1,350 people/day in Dane County, 4,065 people/day in Milwaukee, and 21,680 people/day in NYC); there is a 20% drop in adherence among vaccinated individuals; vaccine effectiveness is 90%; vaccination coverage is 50%; and the baseline test rate is 75%.

## Controllable spread date analysis

Using the implied R0 values for infectious disease models, the herd immunity is theoretically achieved when 70%, 70%, and 85% of the population gains immunization through recovery from COVID-19 or vaccination in Dane County, Milwaukee, and NYC, respectively. However, these herd immunity levels assume that NPIs, including face mask use, are no longer adopted. Therefore, we investigated whether vaccination, along with NPIs, will mitigate the pandemic to a level such that that it will become controllable. For this purpose, we defined a *controllable spread date* as the date after which the number of daily new confirmed cases never exceeds 20 for Dane County, 60 for Milwaukee, and 320 for NYC. For example, if the daily number of new cases never exceeds 20 for Dane County after June 15, 2021, the controllable spread date is set to June 15, 2021. We selected these daily numbers of confirmed cases based on data from Dane County that employs 180 contact tracers to trace COVID-19 cases; therefore, 20 new cases per day would make it feasible to conduct aggressive and efficient contact tracing to completely control the disease in Dane County [14]. For the other regions, we scaled up the number considering population size.

### Sensitivity analysis

We conducted a parametric sensitivity analysis in which we tested the impact of uncertainty in two input parameters: drop in adherence to NPIs after vaccination, where the drop in adherence was assumed to be 0% as opposed to 20% in the base case, and baseline test rate, where the probability of testing in earlier days of the pandemic was assumed to be 50% as opposed to 75% in the base case. We also conducted a structural sensitivity analysis in which we assumed that vaccination reduces the risk of infection instead of leading to complete immunity. That is, a 90% vaccine effectiveness scenario reduces the risk of getting infected among all vaccinated individuals by 90% but does not lead to complete immunity.

## Results

COVAM accurately predicted the reported number of cases in each urban area in the short term (**S2 Fig** in **S1 Data**). We first reported the number of confirmed cases over time for different vaccine coverage and adherence to NPI scenarios when vaccine effectiveness is 90% and daily vaccination capacity is 0.25% (**Fig 1**). We found that the total number of confirmed cases was not sensitive to the vaccination coverage in any of the regions as long as communities keep a high level of adherence to NPIs. In general, the dynamic adherence scenario has led to later controllable spread dates compared to those under fixed adherence scenarios for both vaccination and no vaccination cases.

**Fig 2** and **Table 2** show how adherence to NPIs change the effect of vaccination on the cumulative number of cases. Assuming vaccine effectiveness is 90%, vaccine coverage is 50%, and daily vaccination capacity is 0.25%, we found that the level of adherence to NPIs had a major impact on the cumulative number of cases (**Fig 2** and **Table 2**), as well as the controllable spread date. When the level of NPI adherence is very high (75% for Dane County and Milwaukee; 90% for NYC), the effect of vaccination on the controllable spread date and number of cumulative cases was minimal compared to no vaccination. When the NPI adherence rates are equal to 70% for Dane County and Milwaukee, and 85% for NYC, vaccination reduced the

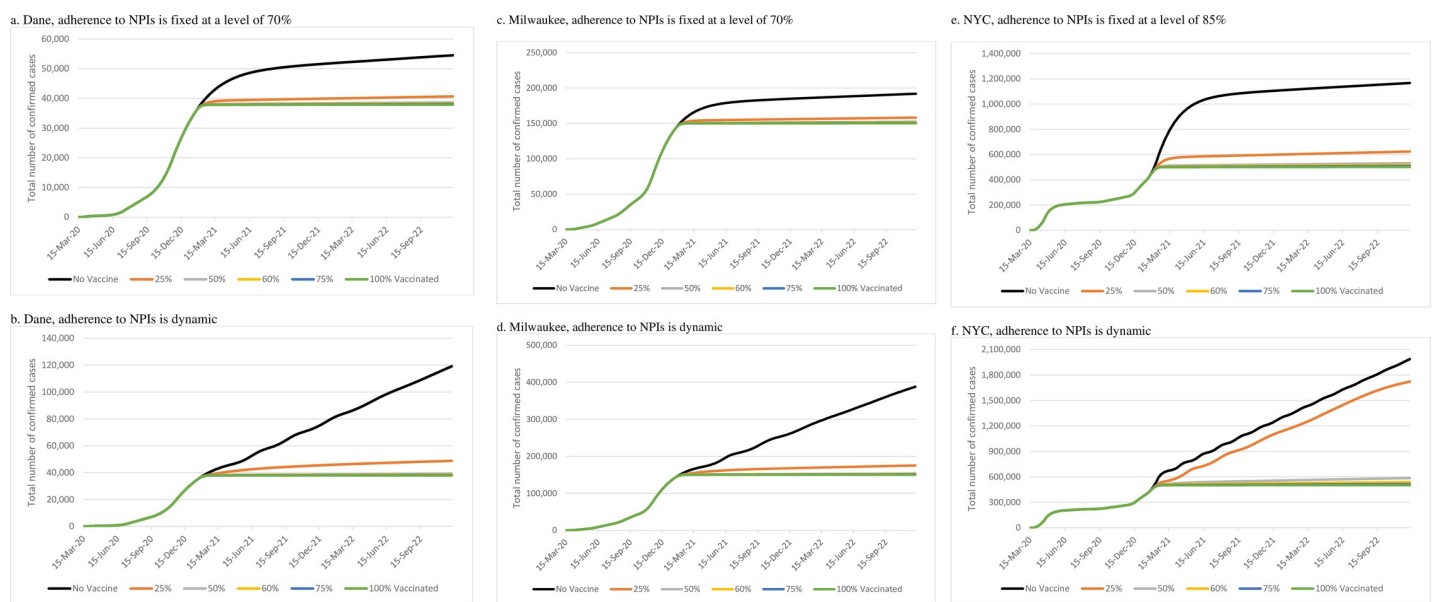

**Fig 1. Impact of vaccine coverage on the number of confirmed cases in different regions for two different scenarios of adherence to nonpharmaceutical interventions (NPIs).**

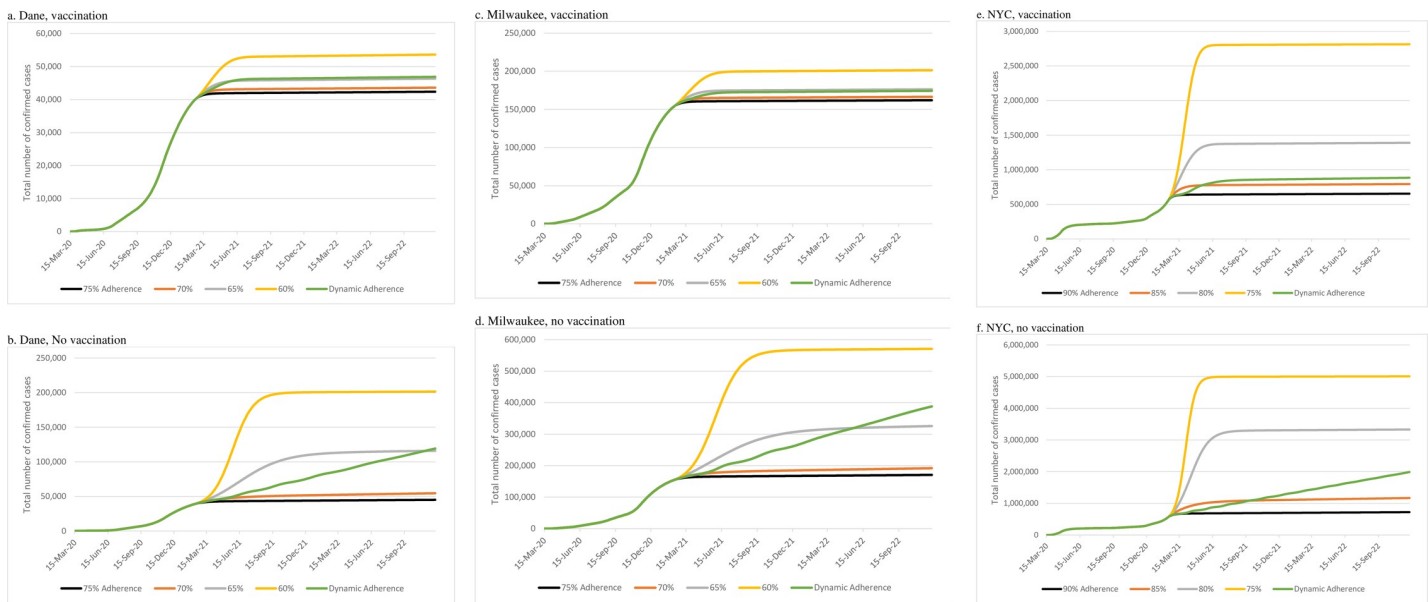

**Fig 2. Impact of vaccination and adherence to nonpharmaceutical interventions (NPIs) on the number of cases over time.**

number of confirmed cumulative cases from 51,784, 185,343, and 1,110,070 to 43,326, 165,679, and 784,137implying an 16%, 11%, and 29% rate of reduction in the number of confirmed cases for Dane, Milwaukee, and NYC, respectively, and reduced the time to controllable spread by 2–3 months in all three regions (**Table 2**, 0.25% vaccination capacity).

The reduction in the number of cases due to vaccination was higher in Dane County compared to Milwaukee. The proportion of cumulative confirmed cases in the population in Dane County and Milwaukee as of January 31, 2021 were 7% and 10%, respectively [15]. COVAM considers the possibility that the actual number of cases is higher than the confirmed number of cases due to some patients experiencing very mild disease and the limited testing. As such, COVAM estimated that the proportion of cumulative confirmed and unreported cases in the population in Dane County and Milwaukee as of January 31, 2021 was 8% and 11%, respectively. Therefore, the prevalence of SARS-CoV-2 was higher in Milwaukee compared to Dane County during the vaccination period.

**Table 2** also shows that maintaining a high level of adherence to NPIs under the same vaccination scenario reduces the number of confirmed cases and results in a significantly earlier date of controllable spread compared to lower NPI adherence in all three regions. For example, in Dane County, a 60% level of NPI adherence after February 1, 2021 led to 53,232 cases with vaccination and a controllable spread date of June 15, 2021 versus a 70% level of NPI adherence, which led to a total of 43,326 cases and a controllable spread date of March 29, 2021.

We also found that daily vaccination capacity had a differential effect on the number of cases and the date on which controllable spread is achieved (**Table 2**). In particular, under the dynamic adherence scenario, vaccination at 0.5% instead of 0.05% of the population per day reduced the number of confirmed cases from 72,062, 250,540, and 1,273,560 to 42,468, 163,134, and 688,177 implying a 41%, 35%, and 46% reduction in Dane County, Milwaukee, and NYC, respectively, and reduced the time to controllable spread by 12, 10, and 16 months, respectively.

**Table 3** shows that the effectiveness of vaccination does not have a major impact on the date when controllable spread is achieved, whereas it reduces the number of confirmed cases greatly. Parametric and structural sensitivity analyses recapitulated the overall trends observed in the base-case runs (**S3-S10 Figs in S1 Data** and **S3-S8 Tables in S1 Data**).

**Table 2. Controllable spread date and number of cases on December 31, 2021 for different daily vaccination capacity and adherence to nonpharmaceutical interventions (NPI) scenarios (vaccine effectiveness 90%, vaccine coverage 50%).**

Dane County

| Vaccination capacity | 75% Adherence | | 70% Adherence | | 65% Adherence | | 60% Adherence | | Dynamic Adherence | |
|---|---|---|---|---|---|---|---|---|---|---|
| | Date | Number of cases | Date | Number of cases | Date | Number of cases | Date | Number of cases | Date | Number of cases |
| No Vaccine | 27-Mar-2021 | 43,932 | 18-Jul-2021 | 51,784 | 17-Mar-2022 | 111,014 | 29-Oct-2021 | 200,642 | After June 2022 | 79,285 |
| 0.05% | 22-Mar-2021 | 43,210 (2%) | 26-May-2021 | 47,977 (7%) | 18-Oct-2021 | 73,339 (34%) | 13-Oct-2021 | 149,440 (25%) | 17-Mar-2022 | 72,062 (9%) |
| 0.1% | 18-Mar-2021 | 42,684 (3%) | 1-May-2021 | 46,063 (11%) | 1-Aug-2021 | 59,141 (47%) | 15-Sep-2021 | 108,001 (46%) | 24-Oct-2021 | 63,856 (19%) |
| 0.25% | 12-Mar-2021 | 42,136 (4%) | 29-Mar-2021 | 43,326 (16%) | 2-May-2021 | 45,999 (59%) | 15-Jun-2021 | 53,232 (73%) | 29-May-2021 | 46,479 (41%) |
| 0.5% | 26-Feb-2021 | 40,698 (7%) | 9-Mar-2021 | 41,477 (20%) | 25-Mar-2021 | 42,819 (61%) | 13-Apr-2021 | 45,326 (77%) | 29-Mar-2021 | 42,468 (46%) |

Milwaukee

| Vaccination capacity | 75% Adherence | | 70% Adherence | | 65% Adherence | | 60% Adherence | | Dynamic Adherence | |
|---|---|---|---|---|---|---|---|---|---|---|
| | Date | Number of cases | Date | Number of cases | Date | Number of cases | Date | Number of cases | Date | Number of cases |
| No Vaccine | 23-Mar-2021 | 167,626 | 12-Jun-2021 | 185,343 | 26-Mar-2022 | 309,809 | 18-Nov-2021 | 567,311 | After June 2022 | 271,692 |
| 0.05% | 18-Mar-2021 | 165,670 (1%) | 7-May-2021 | 177,118 (4%) | 17-Sep-2021 | 229,519 (26%) | 20-Oct-2021 | 416,404 (27%) | 27-Jan-2022 | 250,540 (8%) |
| 0.1% | 14-Mar-2021 | 164,208 (2%) | 19-Apr-2021 | 172,592 (7%) | 13-Jul-2021 | 201,369 (35%) | 8-Sep-2021 | 309,676 (45%) | 30-Sep-2021 | 211,145 (22%) |
| 0.25% | 6-Mar-2021 | 161,278 (4%) | 26-Mar-2021 | 165,679 (11%) | 27-Apr-2021 | 175,355 (43%) | 5-Jun-2021 | 200,362 (65%) | 16-May-2021 | 173,188 (36%) |
| 0.5% | 25-Feb-2021 | 158,520 (5%) | 7-Mar-2021 | 160,586 (13%) | 21-Mar-2021 | 164,078 (47%) | 7-Apr-2021 | 170,184 (70%) | 25-Mar-2021 | 163,134 (40%) |

NYC

| Vaccination capacity | 90% Adherence | | 85% Adherence | | 80% Adherence | | 75% Adherence | | Dynamic Adherence | |
|---|---|---|---|---|---|---|---|---|---|---|
| | Date | Number of cases | Date | Number of cases | Date | Number of cases | Date | Number of cases | Date | Number of cases |
| No Vaccine | 22-Mar-2021 | 699,352 | 27-Aug-2021 | 1,110,070 | 4-Sep-2021 | 3,307,090 | 21-Jun-2021 | 4,999,100 | After June 2022 | 1,308,990 |
| 0.05% | 19-Mar-2021 | 686,109 (2%) | 28-Jun-2021 | 986,660 (11%) | 19-Aug-2021 | 2,758,440 (17%) | 20-Jun-2021 | 4,555,280 (9%) | After June 2022 | 1,273,560 (3%) |
| 0.1% | 17-Mar-2021 | 674,414 (4%) | 30-May-2021 | 909,823 (18%) | 3-Aug-2021 | 2,278,760 (31%) | 18-Jun-2021 | 4,106,650 (18%) | 8-Feb-2022 | 1,186,840 (9%) |
| 0.25% | 10-Mar-2021 | 647,188 (7%) | 22-Apr-2021 | 784,137 (29%) | 14-Jun-2021 | 1,379,770 (58%) | 10-Jun-2021 | 2,808,280 (44%) | 24-Jul-2021 | 862,981 (34%) |
| 0.5% | 3-Mar-2021 | 615,486 (12%) | 27-Mar-2021 | 685,342 (38%) | 28-Apr-2021 | 881,193 (73%) | 3-Jun-2021 | 1,426,150 (71%) | 11-May-2021 | 688,177 (47%) |

Numbers in parentheses represent percent reduction relative to no vaccine.

## Discussion

In this simulation study, we estimated the impact of vaccination on the number of COVID-19 cases in three urban communities using agent-based simulation modeling. We found that controllable spread of SARS-CoV-2 can be achieved sooner than when a large proportion of the population is vaccinated (e.g., 70–80%) as long as there is high adherence to NPIs in the community. We further found that vaccination would reduce the number of COVID-19 cases

**Table 3. Controllable spread date and number of cases on December 31, 2021 for different vaccination effectiveness scenarios (vaccine coverage 50%, vaccination capacity 0.25%).**

Dane County

| Vaccine effectiveness | 75% Adherence | | 70% Adherence | | 65% Adherence | | 60% Adherence | | Dynamic Adherence | |
|---|---|---|---|---|---|---|---|---|---|---|
| | Date | Number of cases | Date | Number of cases | Date | Number of cases | Date | Number of cases | Date | Number of cases |
| No Vaccine | 27-Mar-2021 | 43,932 | 18-Jul-2021 | 51,784 | 17-Mar-2022 | 111,014 | 29-Oct-2021 | 200,642 | After June 2022 | 79,285 |
| 50% | 15-Feb-2021 | 39,421 (10%) | 21-Feb-2021 | 39,835 (23%) | 5-Mar-2021 | 40,620 (63%) | 8-Apr-2021 | 42,721 (79%) | 31-Mar-2021 | 42,166 (47%) |
| 75% | 9-Feb-2021 | 38,632 (12%) | 11-Feb-2021 | 38,808 (25%) | 15-Feb-2021 | 39,070 (65%) | 21-Feb-2021 | 39,540 (80%) | 20-Feb-2021 | 39,486 (50%) |
| 90% | 5-Feb-2021 | 38,267 (13%) | 7-Feb-2021 | 38,373 (26%) | 9-Feb-2021 | 38,519 (65%) | 11-Feb-2021 | 38,742 (81%) | 10-Feb-2021 | 38,719 (51%) |

Milwaukee

| Vaccine effectiveness | 75% Adherence | | 70% Adherence | | 65% Adherence | | 60% Adherence | | Dynamic Adherence | |
|---|---|---|---|---|---|---|---|---|---|---|
| | Date | Number of cases | Date | Number of cases | Date | Number of cases | Date | Number of cases | Date | Number of cases |
| No Vaccine | 23-Mar-2021 | 167,626 | 12-Jun-2021 | 185,343 | 26-Mar-2022 | 309,809 | 18-Nov-2021 | 567,311 | After June 2022 | 271,692 |
| 50% | 15-Feb-2021 | 154,616 (8%) | 20-Feb-2021 | 155,814 (16%) | 2-Mar-2021 | 157,883 (49%) | 27-Mar-2021 | 162,635 (71%) | 18-Mar-2021 | 161,193 (41%) |
| 75% | 8-Feb-2021 | 152,299 (9%) | 11-Feb-2021 | 152,783 (18%) | 14-Feb-2021 | 153,531 (50%) | 20-Feb-2021 | 154,751 (73%) | 19-Feb-2021 | 154,613 (43%) |
| 90% | 5-Feb-2021 | 151,177 (10%) | 7-Feb-2021 | 151,484 (18%) | 8-Feb-2021 | 151,883 (51%) | 10-Feb-2021 | 152,469 (73%) | 10-Feb-2021 | 152,403 (44%) |

NYC

| Vaccine effectiveness | 90% Adherence | | 85% Adherence | | 80% Adherence | | 75% Adherence | | Dynamic Adherence | |
|---|---|---|---|---|---|---|---|---|---|---|
| | Date | Number of cases | Date | Number of cases | Date | Number of cases | Date | Number of cases | Date | Number of cases |
| No Vaccine | 22-Mar-2021 | 699,352 | 27-Aug-2021 | 1,110,070 | 4-Sep-2021 | 3,307,090 | 21-Jun-2021 | 4,999,100 | After June 2022 | 1,308,990 |
| 50% | 23-Feb-2021 | 558,007 (20%) | 15-Mar-2021 | 592,532 (47%) | 7-Jun-2021 | 723,988 (78%) | 17-Dec-2021 | 1,542,430 (69%) | 18-Sep-2022 | 1,150,020 (12%) |
| 75% | 17-Feb-2021 | 527,610 (25%) | 28-Feb-2021 | 542,527 (51%) | 26-Mar-2021 | 578,939 (82%) | 24-Jul-2021 | 718,146 (86%) | 19-Jun-2021 | 627,447 (52%) |
| 90% | 14-Feb-2021 | 513,007 (27%) | 21-Feb-2021 | 521,935 (53%) | 6-Mar-2021 | 539,888 (84%) | 16-Apr-2021 | 588,561 (88%) | 22-Mar-2021 | 557,833 (57%) |

Numbers in parentheses represent percent reduction in the number of cases relative to no vaccine.

significantly, but the rate of reduction in the number of cases differs among regions. Finally, we found that the level of adherence to NPIs has a major impact on the number of cases, as well as the date that controllable spread is achieved regardless of vaccine capacity, vaccine effectiveness, or region. In NYC, vaccination would reduce the number of cases from 3,307,090 to 1,379,770, or by 58%, when the adherence to NPIs is fixed at 80%. Under the same scenario, controllable spread would be achieved in June 2021 as opposed to September 2021 without vaccination.

Although the disease spread rate (i.e., R0 value) was identical for Dane County and Milwaukee, the impact of vaccination on the number of cases and timing of controllable spread was more pronounced in Dane County compared to Milwaukee. Of note, according to the COVAM model, 8% of the Dane County population was infected as of January 31, 2021

compared to 11% of the Milwaukee population. Thus, our results imply that vaccination may have a larger impact when the history of COVID-19 infections is lower to start with. Based on this, we suggest to benefit most from vaccination, regions need to keep a high level of adherence to NPIs to keep caseload low, a similar finding as reported in a prior modeling study [16]. Relying on vaccination alone to control the spread of COVID-19 over the coming months may delay the timing of controllable disease by several months, and this has important implications for the timing of lifting policies for NPIs that are currently in place.

Mainstream media, as well as prior modeling efforts, have focused on the endpoint of herd immunity–the point at which disease transmission is halted and life can "return to normal" from the standpoint of NPIs [17, 18]. While reaching herd immunity is the ultimate goal of a vaccination campaign, we chose to focus on an intermediate metric: controlled spread while continuing reasonable public health precautions. Our results highlight the importance of continued adherence to NPIs during vaccine rollout. As the vaccine is being delivered, our results suggest that vaccine effectiveness in reducing viral spread is highly dependent on NPIs. If NPI efforts are sustained at current levels during the rollout, then even low levels of vaccine uptake will be able to control viral spread. On the other hand, if vaccine rollout is paired with NPI easing, a higher proportion of the population will need vaccination to stop the accumulation of new cases.

Our findings are consistent with those from a few other modeling studies that report reaching herd immunity depends on high adherence to NPIs during vaccination rollout [16, 19]. The impact of even highly effective vaccines may be diminished if deployed into a population with high viral transmission caused by low adherence to NPIs [16]. In addition, the use of NPIs may decrease the level of vaccine coverage needed to achieve herd immunity [19]. Our study extends the existing literature by showing that the degree to which NPIs, vaccine coverage, and pace of vaccination impact disease control may vary regionally, and are highly dependent on both population density and the existing cumulative burden of COVID-19 since the pandemic began. Areas of lower density need lower adherence to NPIs to decrease case rates with vaccination, and those with a lower existing burden of COVID-19 have a greater ability to improve their trajectories with vaccination campaigns.

Our study's findings are also consistent with a few other agent-based simulation models that examined the impact of vaccination on pandemic control [20, 21]. One recent study focused on the pandemic control in the Ontario region in Canada and found that relaxing NPIs and lockdown restrictions early can lead to delayed controllable spread or a second wave of infections [20]. Similarly, we found that dynamic adherence to NPIs was associated with lower infection reduction compared to steady rates of NPI adherence. All vaccination experiments shown in Fig 2 illustrate an additional wave of infections under dynamic adherence to NPIs. Our study also agrees with another study that focused on the US and reported a modest impact of vaccine effectiveness in some cases [21]. Similarly, as shown in Table 3, we also found that greater vaccine effectiveness was not strongly associated with greater reduction in infections.

Our study has several limitations related to uncertainty in vaccine effectiveness in real-world settings. Authorized COVID-19 vaccines to date have demonstrated high efficacy for preventing COVID-19 illness and hospitalization [3]. COVAM assumes that vaccination also prevents transmission of SARS-CoV-2. However, studies to determine the effect of vaccination on acquisition and shedding of SARS-CoV-2 are ongoing. Our model also assumes that COVID-19 vaccines will be effective in preventing transmission of new and future variants of the virus. The B.1.1.7 (UK), B.1.351 (South Africa), and P.1 (Brazil) variants have all been detected in the US [22], and all contain mutations in the spike protein. Both the Pfizer and Moderna vaccines are believed to be effective against the B.1.1.7 variant [23–25]. However, a

recent study suggests the neutralizing antibodies induced by vaccination are less potent against the B.1.351 and B.1.1.7 variants *in vitro* [26]. As long as SARS-CoV-2 is replicating at high levels, new variants can be expected to emerge, underscoring the importance of high-level adherence to NPIs and rapid vaccination with high-level coverage. Furthermore, we do not use a full calibration procedure that is commonly used in simulation modeling to estimate the unobservable inputs of the model, which may have led to suboptimal set of inputs [27–29]. Finally, our model does not consider age-based differences in the administration and effectiveness of the vaccines. However, currently used vaccines are administered for individuals over ages 16 only. It is unknown if vaccination of individuals younger than 16 of age will begin by the end of the study's simulation period or if the effectiveness will be different for younger individuals.

In conclusion, this simulation modeling demonstrates that continued high adherence to NPIs along with vaccination results in a shorter time to control the COVID-19 pandemic in US urban areas. Furthermore, our results suggest adhering to NPIs to keep the caseload low until vaccine becomes widely available can lead to greater benefit of vaccination in terms of reduction in the number of COVID-19 cases.

## Supporting information

**S1 Data. Supplementary material.**
(DOCX)

## Acknowledgments

The authors thank Amanda Young for her help in preparing this manuscript.

## Author Contributions

**Conceptualization:** Oguzhan Alagoz, Ajay K. Sethi, Brian W. Patterson, Matthew Churpek, Nasia Safdar.

**Formal analysis:** Oguzhan Alagoz, Ghalib Alhanaee, Nasia Safdar.

**Funding acquisition:** Nasia Safdar.

**Investigation:** Oguzhan Alagoz, Ajay K. Sethi, Brian W. Patterson, Elizabeth Scaria, Nasia Safdar.

**Methodology:** Oguzhan Alagoz, Matthew Churpek, Ghalib Alhanaee, Elizabeth Scaria.

**Project administration:** Oguzhan Alagoz.

**Software:** Oguzhan Alagoz, Ghalib Alhanaee.

**Supervision:** Oguzhan Alagoz, Nasia Safdar.

**Validation:** Oguzhan Alagoz.

**Visualization:** Oguzhan Alagoz.

**Writing – original draft:** Oguzhan Alagoz, Ajay K. Sethi, Brian W. Patterson, Matthew Churpek, Elizabeth Scaria, Nasia Safdar.

**Writing – review & editing:** Oguzhan Alagoz, Ajay K. Sethi, Brian W. Patterson, Matthew Churpek, Ghalib Alhanaee, Elizabeth Scaria, Nasia Safdar.

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
