## [Decision Letter · Decision Letter 0]

11 May 2021

PONE-D-21-10365

The Impact of Vaccination to Control COVID-19 Burden in the United States: A Simulation Modeling Approach

PLOS ONE

Dear Dr. Alagoz,

Thank you for submitting your manuscript to PLOS ONE. After careful consideration, we feel that it has merit but does not fully meet PLOS ONE’s publication criteria as it currently stands. Therefore, we invite you to submit a revised version of the manuscript that addresses the points raised during the review process.

We look forward to receiving your revised manuscript.

Kind regards,

Martin Chtolongo Simuunza, PhD

Academic Editor

PLOS ONE

Journal Requirements:

2. In your ethics statement in the Methods section and in the online submission form, please provide additional information about the data used in your retrospective study. Specifically, please ensure that you have discussed whether all data were fully anonymized before you accessed them and/or whether the IRB or ethics committee waived the requirement for informed consent. If patients provided informed written consent to have data from their medical records used in research, please include this information.

"OA is a paid consultant for Bristol Myers Squibb and Biovector Inc, outside of the submitted work. Other authors have no conflicts of interest to report."

Reviewers' comments:

Reviewer's Responses to Questions

**Comments to the Author**

1. Is the manuscript technically sound, and do the data support the conclusions?

Reviewer #1: Yes

Reviewer #2: Yes

2. Has the statistical analysis been performed appropriately and rigorously? 

Reviewer #1: Yes

Reviewer #2: Yes

3. Have the authors made all data underlying the findings in their manuscript fully available?

Reviewer #1: Yes

Reviewer #2: No

4. Is the manuscript presented in an intelligible fashion and written in standard English?

Reviewer #1: Yes

Reviewer #2: Yes

5. Review Comments to the Author

Reviewer #1: This paper examines the impact of COVID-19 vaccination in the context of current nonpharmaceutical interventions in three urban jurisdictions in the United States. A strength of their agent-based model is the detailed parameterization of interactions and representation of varying levels of adherence to NPIs. Overall, the paper is well written.

1. Please update the introduction to reflect the current situation with COVID-19 vaccination in the United States.

2. In Table 1, the parameterization of vaccine effectiveness should be updated based on recent literature and data reported in real-world studies. Do you distinguish between vaccine efficacy against infection, symptomatic disease, and severe disease?

3. What is the timeline between the first and second doses? Do you account for changes in vaccine efficacy after the first and second doses, respectively?

4. The base case scenario assumes a daily vaccination capacity of 0.1% of the population per day for Dane County, Milwaukee, and NYC. Can this be supported by current vaccination roll-out data in these 3 regions? Do you anticipate this to change in the future with vaccine hesitancy?

5. Does the model account for pre-existing immunity (i.e., from previous COVID-19 infection)?

6. It is suggested in the paper that regions need to maintain a high level of adherence to NPIs to keep caseloads low. At what point in the vaccination campaign can these NPI measures be lifted successfully?

7. A number of other agent-based modelling papers have already been published regarding the population-level impact of COVID-19 vaccination in various jurisdictions (see below). Please situate your findings in the context of the existing literature.

https://doi.org/10.1093/cid/ciab079

https://doi.org/10.1016/j.vaccine.2021.03.058

Reviewer #2: The paper presents an agent-based model integrating adherence to NPIs, vaccination strategies (and possibilities) and the interactions between both of them.

The current paper presents an extension of an already existing model (COVAM), adding vaccination strategies.

A first point is that the paper is not self-content as far as the presentation of the model is concerned: the description is too short to understand what are the characteristics and behaviors of the agents. The description makes the reader question the fact that the model is really an agent-based model and not a micro-simulation model.

p4: "The objective of this study was to use our previously developed agent-based simulation model (2),"

it could be clearer to express that this previously developed model is a COVID-19 model (a lot of COVID models are adaptation of models about other diseases.

Figure s1. Progression of COVID-19 (adapted from Alagoz, et al 2020(1))

The epidemiological model has some flaws:

- There is no compartiment for asymptomatic, which is a key feature of the COVID-19

- if IM- is undetected and IM+ detected, it seems intuitive to have an arrow from IM- to IM+

- Instead of infected, infectious would more reflect what is important from an epidemiological point of view.

Methods :

Nothing is said about experimental conditions, in particular concerning the simulations launches conditions: use of HPC or personal computers ...

page9: "We ran 100 replications for each experiment" : why this number of 100 replications, how is it justified.

About adherence

"For instance, a 70% adherence level to NPIs in Dane County and Milwaukee is implemented by simply reducing the number of daily contacts from 10 per day to 3 per day per person, which slows the transmission of SARS-CoV-2."

Even if an individual has a 100% adherence level to NPIs, its daily contacts will not sink to 0... (it can be in family, it will need to buy some essential goods (foods...)).

This idea of adherence to NPIs is interesting, but its implementation seems to simplistic.

6. PLOS authors have the option to publish the peer review history of their article (what does this mean?). If published, this will include your full peer review and any attached files.

Reviewer #1: No

Reviewer #2: No

---

## [Author Response · Author response to Decision Letter 0]

14 May 2021

We have attached a separate point-by-point response document.

---

## [Decision Letter · Decision Letter 1]

28 Jun 2021

The Impact of Vaccination to Control COVID-19 Burden in the United States: A Simulation Modeling Approach

PONE-D-21-10365R1

Dear Dr. Alagoz,

We’re pleased to inform you that your manuscript has been judged scientifically suitable for publication and will be formally accepted for publication once it meets all outstanding technical requirements.

Kind regards,

Martin Chtolongo Simuunza, PhD

Academic Editor

PLOS ONE

Additional Editor Comments (optional):

Reviewers' comments:

Reviewer's Responses to Questions

**Comments to the Author**

1. If the authors have adequately addressed your comments raised in a previous round of review and you feel that this manuscript is now acceptable for publication, you may indicate that here to bypass the “Comments to the Author” section, enter your conflict of interest statement in the “Confidential to Editor” section, and submit your "Accept" recommendation.

Reviewer #1: All comments have been addressed

Reviewer #2: All comments have been addressed

2. Is the manuscript technically sound, and do the data support the conclusions?

Reviewer #1: Yes

Reviewer #2: Yes

3. Has the statistical analysis been performed appropriately and rigorously? 

Reviewer #1: Yes

Reviewer #2: Yes

4. Have the authors made all data underlying the findings in their manuscript fully available?

Reviewer #1: Yes

Reviewer #2: Yes

5. Is the manuscript presented in an intelligible fashion and written in standard English?

Reviewer #1: Yes

Reviewer #2: Yes

6. Review Comments to the Author

Reviewer #1: (No Response)

Reviewer #2: (No Response)

7. PLOS authors have the option to publish the peer review history of their article (what does this mean?). If published, this will include your full peer review and any attached files.

Reviewer #1: No

Reviewer #2: No

---

## [Editor Report · Acceptance letter]

6 Jul 2021

PONE-D-21-10365R1 

The Impact of Vaccination to Control COVID-19 Burden in the United States: A Simulation Modeling Approach 

Dear Dr. Alagoz:

I'm pleased to inform you that your manuscript has been deemed suitable for publication in PLOS ONE. Congratulations! Your manuscript is now with our production department. 

Kind regards, 

on behalf of

Dr. Martin Chtolongo Simuunza 

Academic Editor

PLOS ONE